# Grass–Legume Mixture with *Rhizobium* Inoculation Enhanced the Restoration Effects of Organic Fertilizer

**DOI:** 10.3390/microorganisms11051114

**Published:** 2023-04-25

**Authors:** Haijuan Zhang, Kaifu Zheng, Songsong Gu, Yingcheng Wang, Xueli Zhou, Huilin Yan, Kun Ma, Yangan Zhao, Xin Jin, Guangxin Lu, Ye Deng

**Affiliations:** 1College of Agriculture and Animal Husbandry, Qinghai University, Xining 810016, China; 15297196583@163.com (H.Z.); zhengkf1986@163.com (K.Z.); yingcheng_w@163.com (Y.W.); zhouxuelia@163.com (X.Z.); yan06112021@163.com (H.Y.); m18294258973@163.com (K.M.); 18893093541@163.com (Y.Z.); 18894310895@163.com (X.J.); 2Research Center for Eco-Environmental Sciences, Chinese Academy of Sciences, Beijing 100085, China; 3Experimental Station of Grassland Improvement in Qinghai Province, Gonghe 813000, China; 4College of Resources and Environment, University of Chinese Academy of Sciences, Beijing 100190, China

**Keywords:** artificial grassland, Qinghai–Tibet Plateau, grassland yield, soil nutrient, native mycorrhizal fungi

## Abstract

The establishment of artificial grassland is crucial in restoring degraded grassland and resolving the forage–livestock conflict, and the application of organic fertilizer and complementary seeding of grass–legume mixture are effective methods to enhance grass growth in practice. However, its mechanism behind the underground is largely unclear. Here, by utilizing organic fertilizer in the alpine region of the Qinghai–Tibet Plateau, this study assessed the potential of grass–legume mixtures with and without the inoculation of *Rhizobium* for the restoration of degraded grassland. The results demonstrated that the application of organic fertilizer can increase the forage yield and soil nutrient contents of degraded grassland, and they were 0.59 times and 0.28 times higher than that of the control check (CK), respectively. The community composition and structure of soil bacteria and fungi were also changed by applying organic fertilizer. Based on this, the grass–legume mixture inoculated with *Rhizobium* can further increase the contribution of organic fertilizer to soil nutrients and thus enhance the restoration effects for degraded artificial grassland. Moreover, the application of organic fertilizer significantly increased the colonization of gramineous plant by native mycorrhizal fungi, which was ~1.5–2.0 times higher than CK. This study offers a basis for the application of organic fertilizer and grass–legume mixture in the ecological restoration of degraded grassland.

## 1. Introduction

The natural grassland on the Qinghai–Tibet Plateau is a significant production base for grassland livestock husbandry in China [1], accounting for nearly one-third of China’s grassland area [2]. However, the natural grassland has been severely degraded due to global climate change and excessive human production activities, such as overgrazing, engineering operations, and insect pests, resulting in a decrease in grassland productivity and soil fertility. The establishment of artificial grassland is crucial in resolving the forage–livestock conflict and relieving the pressure on natural grassland [3,4]. However, the stability of the established artificial grassland is weak due to the planting patterns and lack of high-quality native grass species, making it challenging to maintain the sustainability of production and ecological services without field conservation and management practices such as fertilization and complementary seeding [5].

Fertilization is an effective measure for maintaining soil nutrient balance and increasing grassland productivity [5]. Chemical fertilizer is primarily utilized due to its efficiency, which can promote the rapid growth of grass within a short period. However, the long-term overuse of chemical fertilizers will result in increasing pollution and soil hardening in alpine regions [6,7], inevitably decreasing the productivity of grassland [8,9]. Therefore, environmentally friendly organic fertilizer has been applied for the ecological restoration of degraded grassland due to its high fertilizer efficiency and comprehensive nutrition [10]. A previous study had shown that soil nutrients and the microbial community of degraded alpine grassland can be quickly improved by adding organic carbon and nitrogen [11]. However, the degradation of vegetation and soil on the Qinghai–Tibet Plateau cannot be completely mitigated by fertilization alone; complementary seeding and other measures are also required.

Complementary seeding can increase the proportion of dominant grasses in degraded grassland [12]. According to prior studies, the restoration effect of complementary seeding based on a single grass species was not stable, but the stability was significantly enhanced after increasing the number of grass species. A grass–legume mixture is considered an efficient planting system as a result of niche complementarity [13], which can increase grassland yield and soil fertility [14]. *Rhizobium* can induce the cortical cells of roots and stems of legumes to proliferate, resulting in the formation of root nodules [15], within which they fix atmospheric nitrogen into ammonia, which is mostly supplied for the growth of legumes, reducing the need for extra input of nitrogen [16,17]. Additionally, legumes share excess nitrogen with gramineous grasses in a variety of ways [18,19]. It is reported that alfalfa seeds inoculated with *Rhizobium* can improve grass yield and nodule number [20]. Therefore, inoculating with *Rhizobium* before seeding is a vital step to promote the yield of legumes.

Previous studies evaluated the restoration effects of degraded grassland mainly based on vegetation and soil physico-chemical properties while ignoring the critical role of soil microorganisms, which directly or indirectly affect the structure and function of grassland ecosystems by participating in multiple ecological processes such as biological nitrogen fixation and litter decomposition [21,22] and are more sensitive to external environmental changes [23]. In recent years, an increasing number of researchers have begun to explore the mechanisms behind the ecological restoration of degraded grassland in terms of soil microbial diversity, community structure, and functions. The findings indicated that (i) the restoration of degraded grassland was closely related to microbial activities, and (ii) the application of organic fertilizer changed the species composition, structure, richness, and diversity of bacterial and fungal communities in the soil [24], (iii) facilitated the enrichment of beneficial microorganisms [25] and (iv) strengthened the colonization of arbuscular mycorrhizal fungi, a significant symbiotic microbial group in the soil, to gramineous plants [26]. Furthermore, the grass–legume mixture increased the plant yield while enhancing the stability and complexity of the rhizosphere soil bacterial community network. The higher the stability and complexity, the greater the contribution to the forage growth [27]. These results demonstrated that the microbial communities will react to fertilization and complementary seeding, and it will feed back to the vegetation or soil of degraded grassland by altering the community structure or network stability, which will enhance the restoration effects of the degraded grassland. However, such research is still uncommon in alpine regions. Strengthening research on the reaction of soil microbial communities to restoration strategies is important to support the ecological restoration of degraded grassland in the alpine region on the Qinghai–Tibet Plateau. Therefore, this study explored the effects of applying organic fertilizer and complementary seeding of grass–legume mixtures on the yield, soil nutrients, and microbial communities of a degraded artificial grassland in the alpine region as well as the colonization rate of native arbuscular mycorrhizal fungi (AMF) on a perennial grass of the Gramineae family. The results can provide some references for the restoration of degraded grassland on the Qinghai–Tibet Plateau.

## 2. Materials and Methods

### 2.1. Site Overview and Study Design

The study site was situated within the test area of artificial grassland of the Experimental Station of Grassland Improvement in Qinghai Province (37°03’ N, 99°33’ E; 3270 m. Figure 1). Meteorological data show that the average annual temperature of the test area is −0.7 °C, July is the warmest month with an average temperature of 17.5 °C and January is the coldest month with an average temperature of −22.6 °C. The average annual precipitation is 368.1 mm, which is mainly concentrated in June, July, and August, and the average annual evaporation is 1495.3 mm. In 2016, the Experimental Station of Grassland Improvement in Qinghai Province planted *Poa pratensis* ‘Qinghai’ in this area as an artificial grassland and used it as a seed field for three years, with seeds being harvested once a year and weeded with 2,4-D butyl ester. In 2019, *Elymus breviaristatus* was planted in the field after the seed collection of *Poa pratensis* ‘Qinghai’, and the grassland was closed for dry farming. Without any field management practice, it was observed that the vegetation coverage and the dominant grasses decreased, and poisonous weeds increased. By the middle of June 2020, before the experiment was conducted, it was a degraded artificial grassland with *P. pratensis* ‘Qinghai’ and *E. breviaristatus* as the dominant species, along with *Plantago asiatica*, *Potentilla anserina*, *Astragalus membranaceus*, *Artemisia scoparia*, *Elsholtzia densa*, and other forbs, and the soil type was dark chestnut soil.

In June 2020, two sample regions with identical conditions were chosen in the degraded artificial grassland and divided into the organic fertilizer group (F) and the no organic fertilizer group (both with an area of 0.25 hm^2^, Figure 1). Then, within each sample region, 15 plots, each with an area of 15 m^2^ (3 × 5 m), were randomly placed, totaling 30 plots. For both groups, 1.5 m wide buffer strips were set between each plot, and these buffer strips were left untreated. For the organic fertilizer group, 50 kg/ha of organic fertilizer was manually and evenly applied. The organic fertilizer was purchased from the Pujiang Greenland Straw Science and Technology Co., Ltd., the technical partner of the College of Resources, Sichuan Agricultural University, with an organic matter content ≥ 45%, N + P_2_O_5_ + K_2_O ≥ 5%, moisture ≤ 30%, mainly composed of organic-rich crop straw and poultry manure as raw materials with the addition of a biological starter. For the no organic fertilizer group, no fertilization was provided. After fertilization, a randomized block experiment was conducted on the 15 plots of each group, respectively, including no seeding (CK), complementary seeding of perennial gramineous plant *Elymus sibiricus* ‘Chuancao No 2′ and leguminous plant *Medicago sativa* ‘Beilin 202′ (R), and complementary seeding of *E. sibiricus* ‘Chuancao No 2′ and *M. sativa* ‘Beilin 202′ inoculated with *Rhizobium* (IR); each treatment was replicated 5 times. *Rhizobium* used in this study was a concentrated agent purchased from Shanghai Ruichu Biotechnology Co., Ltd. The inoculation method was carried out by adding 500 mL of the *Rhizobium* agent to 25 kg seeds of *M. sativa* ‘Beilin 202′, which was followed by mixing and air drying. The seeds inoculated with *Rhizobium* were sown within 12 h. The seeds for testing were provided by the Experimental Station of Grassland Improvement in Qinghai Province. The seed quantities of *E. sibiricus* ‘Chuancao No 2′ and *M. sativa* ‘Beilin 202′ were 22.5 g/15 m^2^ and 45 g/15 m^2^, respectively, which were manually and uniformly sown by drilling, and then, they were closed for dry farming.

### 2.2. Vegetation and Soil Sampling

The collection of vegetation and soil samples was completed in August 2020. Five quadrats, each 50 × 50 cm, were randomly located within each plot (15 m^2^), within which the total vegetation coverage was measured and recorded as average height (10 replications). The fresh weight of all aboveground parts of all plants in the quadrats (ground cutting) were also measured. Then, a soil auger with an inner diameter of 5 cm was used to collect 3 fresh soil samples at a depth of 0–15 cm in the quadrats, which were placed in the numbered self-sealing bags and immediately stored in a refrigerated (4 °C) transport box. The soil temperature, moisture, and conductivity of each plot were measured randomly 15 times using a FieldScout (Spectrum Technologies TDR 350, USA). Furthermore, the entire root system of *E. breviaristatus* was collected to determine the native mycorrhizal colonization rate (5 plants were harvested, repeated 3 times). Soil samples were homogenized and divided into two parts: one for the determination of soil nutrient contents (stored at 4 °C) and the other for soil microorganisms (stored at −80 °C).

### 2.3. Determination of Soil Nutrient Contents

The soil nutrient status was assessed by the contents of soil organic matter, total nitrogen, ammonia nitrogen, and nitrate nitrogen. Soil organic matter content was calculated multiplying soil organic carbon content by a factor of 1.724, and the measurement of soil organic carbon was referred to Walkley et al. [28]. Soil total nitrogen content was determined by Kjeldahl [29]. Soil ammonia nitrogen and nitrate nitrogen content were extracted with 2 M potassium chloride solution, and then, we measured the absorbance of the supernatant after centrifugation at 420 nm and 210 nm, respectively [30].

### 2.4. Determination of Native Mycorrhizal Colonization Rate

Whole roots of *E. breviaristatus* were cleaned with tap water carefully and divided into segments of 1–2 cm before being placed in a centrifuge tube containing formaldehyde—acetic acid—alcohol (FAA) for overnight fixing. The next day, the native mycorrhizal colonization rate was determined using the method by Blažková et al. [31], and the mycorrhizal structure was identified by the Trypan Blue staining method [32]. The specific steps were as follows: First, an appropriate amount of plant root segments was taken from the FAA fixative, and filter paper was used to absorb water after rinsing them three times. These segments were then placed in a 10 mL centrifuge tube and incubated for 1 h with 5 mL of 10% KOH solution in a 90 °C water bath. After discarding the excess KOH solution, the root segments were rinsed under running water once. Then, 5 mL of alkaline hydrogen peroxide (30 mL of 10% H_2_O_2_, 3 mL of NH_4_OH, 567 mL of distilled water) was added, and the root segments were bleached for 15 min at room temperature until they turned white. After bleaching and being rinsed with water, 5 mL of 5% lactic acid was added for 5 min to acidify the samples. After the acidification and subsequent rinsing with water, 5 mL of 0.05% trypan blue dye (0.05 g of trypan blue, 100 mL of lactic acid, 100 mL of glycerin, 100 mL of distilled water) was added, and the root segments were incubated at 90 °C for 30 min. The excess dye solution was discharged, and the root samples were rinsed with water. Next, 5 mL of lactic acid glycerin (50 mL of lactic acid, 100 mL of glycerin, 50 mL of distilled water) was added to decolorize the roots overnight at room temperature. The next day, 5 mL of lactic acid glycerin was added again after discarding the overnight incubation solution. Glass slides and coverslips were wiped with 75% of alcohol and placed on a flat workbench. Tweezers were used to choose 10 root segments with approximately the same thickness; these root segments were placed on the glass slide in order. After adding 1 drop of lactic acid and covering the segments with the coverslip, the segments were gently squeezed until they were flattened, and any bubbles were removed. The excess lactic acid at the edge was wiped away, and the native mycorrhizal structure under the optical microscope (10× mirror) was observed and recorded through photos 3 times.

### 2.5. DNA Extraction and PCR Amplification

Total DNA was extracted from a 0.25 g of soil sample using the Power soil™ DNA Isolation Kit (MO BIO Laboratories, Carlsbad, CA, USA), following the manufacturer’s instructions. After the extracted DNA passed the quality inspection (concentration was above 20 ng/μL, A260/A280 was 1.8–2.0), PCR amplification was carried out, using the universal primers 515 forward (5′-GTGCCAGCMGCCGCGGTAA-3′) and 806 reverse (5′-GGACTACHVGGGTWTCTAAT-3′) for the V4 region of the 16S rRNA genes from bacteria and the primers 5.8 forward (5′-AACTTTYRRCAAYGGATCWCT-3′) and 4 reverse (5′-AGCCTCCGCTTATTGATATGCTTAART-3′) in the ITS2 region for fungi. The reaction system of PCR amplification included 25 μL of 10 × PCR buffer, 20 μL of ddH_2_O, 1 μL of dNTP mixture, 1 μL of Taq DNA Enzyme (TaKaRa), 1 μL 10 uM forward primer, 1 μL of 10 uM reverse primer, and 1 μL of template DNA (20–30 ng/μL). The reaction procedures were as follows: initial denaturation at 95 °C for 3 min, 30 (16S)/35 (ITS) cycles at 95 °C for 15 s, 15 s at 52 °C, 45 s at 72 °C, and a final extension at 72 °C for 5 min, which was held at 4 °C. The PCR products were purified on a 1% agarose gel and then further purified using the E.Z.N.A.™ Gel Extraction Kit (Omega Bio-tek, Inc., Norcross, GA, USA). After which, a Qubit fluorometer (Life technologies Holdings Pte Ltd., Singapore) was employed for quantitative studies, a VAHTSTM Nano DNA Library Prep Kit for Illumina^®^ (Vazyme Biotech Co., Ltd., Nanjing, China) was used for library construction, and the purified amplicons were sequenced on the Illumina HiSeq platform at Magigene Biotechnology Co., Ltd. (Guangzhou, China). 

### 2.6. Sequencing Data Processing

Raw reads of the 16S and ITS genes obtained by sequencing were processed using a publicly available pipeline which was integrated with a variety of bioinformatics tools (http://mem.rcees.ac.cn: 8080, accessed on 20 April 2023) [33,34]. The raw reads were assigned to each sample according to the barcodes, with one mismatch allowed. The Flash program was used to combine the forward and reverse reads of the same sequence [35], and the reads were then filtered using the Btrim program [36] to ensure that the quality score threshold was larger than 20, and the minimum length was 140 bp for 16S and 300 bp for ITS. Following that, UPARSE was utilized to remove the chimeras and to cluster the operational taxonomic units (OTUs) with a similarity of 97% to generate the OTU table [37]. The generated OTU table was flattened to obtain the Resample OTU table to correct the sampling error and ensure downstream analyses with equal sequencing depth (245/307 bp for 16S, 260/341 bp for ITS).

### 2.7. Data Analysis

Microsoft Excel 2010 was used to input and sort out all test data. IBM SPSS Statistics 26 was used for one-way ANOVA, and a *t*-test was used to determine the statistical differences in correlation factors of vegetation, soil, and microorganisms as well as native mycorrhizal colonization rate (*p <* 0.05). The α-diversity of soil microorganisms was assessed by the Chao 1 index and Simpson_evenness index, and the structural difference between soil bacterial and fungal communities was characterized by the results of the permutation multivariate analysis of variance (Per-MANOVA/Adonis) test based on the Bray–Curtis distance. OTUs of soil bacteria and fungi were annotated in the Greengene and Unite databases, respectively, and species components were compared at the phylum and genus levels. The Mantel test was used to confirm the influence intensity of vegetation and soil factors on soil bacterial and fungal communities. Origin 2022 was employed to draw box plots, stacked bar charts, and bar charts. The charts of correlation were drawn using the platform of Lingbo MicroClass (http://www.biomicroclass.com/, accessed on 20 April 2023), and the native mycorrhizal structures were photographed using an optical microscope (10×).

## 3. Results

### 3.1. Increased Yield and Soil Nutrient Contents Due to Organic Fertilizer

As shown in Table 1, compared with the no organic fertilizer group (CK), the application of organic fertilizer (F) significantly enhanced the vegetation coverage (*p <* 0.05), and the contents of soil organic matter, total nitrogen, nitrate nitrogen, and ammonia nitrogen also increased to varying degrees.

After the complementary seeding of *E. sibiricus* ‘Chuancao No 2′ and *M. sativa* ‘Beilin 202′ based on the application of organic fertilizer (FR), the contents of soil total nitrogen, nitrate nitrogen and organic matter increased, but the contents of soil ammonia nitrogen and vegetation coverage decreased. However, there was no statistical difference when compared with the organic fertilizer group (FR-vs-F).

The contents of soil total nitrogen, ammonia nitrogen, nitrate nitrogen and vegetation coverage were all increased after carrying out *Rhizobium* inoculation on *M. sativa* ‘Beilin 202′ in the complementary seeding group (FIR), while the content of soil organic matter exhibited a decreased trend (FIR-vs-FR). However, except for the content of nitrate nitrogen, there was no significant difference compared with the organic fertilizer group (FIR-vs-F). 

### 3.2. Organic Fertilizer Application Influenced Soil Microbial Diversity and Species Composition

Compared with the no organic fertilizer (CK), the application of organic fertilizer (F) increased the Chao 1 index and Simpson_evenness index of soil bacteria in degraded artificial grassland, and the effect was enhanced after complementary seeding based on the application of organic fertilizer (FR) (Figure 2a,b). However, after carrying out *Rhizobium* inoculation on *M. sativa* ‘Beilin 202′ (FIR), the Simpson_evenness index of soil bacteria showed a downward trend. Organic fertilizer (F), on the other hand, increased the Chao 1 index and decreased the Simpson_evenness index of soil fungi. The Chao 1 index was significantly increased after complementary seeding (FR) (*p* < 0.01), and there was also a slight increase in the Simpson_evenness index compared with the application of organic fertilizer (F). After carrying out *Rhizobium* inoculation on *M. sativa* ‘Beilin 202′ (FIR), the Chao 1 index of soil fungi showed a downward trend (Figure 2c,d).

As shown in Table 2, the results of the permutational multivariate analysis of variance based on the Bray–Curtis distance (Adonis) demonstrated that the application of organic fertilizer significantly affected the community structure of soil bacteria and fungi in degraded artificial grassland (*p <* 0.05). However, the effects of complementary seeding alone and complementary seeding inoculated with *Rhizobium* on the bacterial and fungal community structure were not significant, but the impact of complementary seeding inoculated with *Rhizobium* was higher than that of complementary seeding. The impact of organic fertilizer, at the same time, was higher on the community structure of fungi than bacteria.

As shown in Figure 3a, the application of organic fertilizer (F) increased the soil bacterial abundance of *Proteobacteria*, *Bacteroidetes,* and *Firmicutes* of degraded artificial grassland. However, the abundance of *Firmicutes* decreased after conducting complementary seeding (FR) or complementary seeding inoculated with *Rhizobium* based on the application of organic fertilizer (FIR). Moreover, the application of organic fertilizer (F) decreased the abundance of *Actinobacteria, Acidobacteria, Gemmatimonadetes, Planctomycetes, Thaumarchaeota* and *Verrucomicrobia,* and it significantly decreased the abundance of *Chloroflexi* (*p <* 0.05). After complementary seeding based on the application of organic fertilizer (FR), the abundance of *Actinobacteria* exhibited a significant decrease (*p <* 0.05). 

As shown in Figure 3c, the application of organic fertilizer (F) increased the abundance of *Sphingomonas* and *Pseudarthrobacter*, but there was no significant change after conducting complementary seeding (FR) or complementary seeding inoculated with *Rhizobium* (FIR). As for soil fungi, the application of organic fertilizer (F) increased the abundance of *Basidiomycota* and *Glomeromycota*, and it decreased the abundance of *Ascomycota* (*p <* 0.05)*, Chytridiomycota,* and *Zygomycota*, as shown in Figure 3b. After complementary seeding based on the application of organic fertilizer (FR), the abundance of *Basidiomycota* decreased. 

As shown in Figure 3d, the application of organic fertilizer (F) increased the abundance of *Preussia*, *Gibberella*, *Fusarium*, and *Lophiostoma* and decreased the abundance of *Entoloma*, *Podospora*, *Pyrenophora*, *Aleuria*, *Psathyrella*, *Thelebolus*, *Conocybe*, *Embellisia*, *Trechispora*, *Exophiala*, and *Pseudogymnoascus*. After complementary seeding (FR) or complementary seeding inoculated with *Rhizobium* (FIR) based on the application of organic fertilizer, the positive impact of organic fertilizer on the abundance of *Preussia* and *Lophiostoma* decreased, but the abundances of *Aleuria*, *Conocybe*, and *Trechispora* increased.

### 3.3. Correlation between Environmental Factors and Soil Microbial Community Structure

To assess the influence of environmental factors on soil bacterial and fungal communities of degraded artificial grassland, the Mantel test was conducted on the correlation between vegetation coverage (VC), vegetation height (VH), vegetation aboveground biomass (VB), soil total nitrogen (TN), ammonia nitrogen (AN), nitrate nitrogen (NN), organic matter (SOM), soil temperature (ST), moisture (SM), conductivity (SC) and the structure, α-diversity, and abundance of dominant genera of soil bacterial and fungal communities (Figure 4). The results demonstrated that there was no influence by the environmental factors on the structure, α-diversity, and abundance of dominant genera of soil bacterial communities, but soil nitrate nitrogen contents significantly influenced the α-diversity of soil fungi (*p* < 0.05).

### 3.4. Increased Native Mycorrhizal Colonization Rate Due to Organic Fertilizer

The native mycorrhizal colonization rate was assessed in the roots of *E. breviaristatus* from each treatment group. The findings revealed that vesicles, hyphae, and other structures can be found in each treatment group (Figure 5a). Compared with the no organic fertilizer group (CK) (19.33% ± 1.53), the organic fertilizer group (F) had the highest native mycorrhizal colonization rate (59% ± 2.00), which was followed by the organic fertilizer + grass–legume mixture group (FR) (53% ± 2.00), organic fertilizer + grass–legume mixture inoculated with *Rhizobium* group (FIR) (50.67% ± 2.52), grass–legume mixture inoculated with *Rhizobium* group (IR) (36.67% ± 3.06), and grass–legume mixture group (R) (27.33% ± 3.06) (Figure 5b). The findings also revealed that the native mycorrhizal colonization rate of the organic fertilizer group (F) was significantly higher than that of the CK group (*p* < 0.001), as was the FR group over the R group (*p* < 0.001) and the FIR group over the IR group (*p* < 0.001). Based on these results, it can be concluded that the application of organic fertilizer significantly increased the colonization of native mycorrhizal fungi on perennial gramineous forage in degraded artificial grassland, whereas a grass–legume mixture or grass–legume mixture inoculated with *Rhizobium* based on the application of organic fertilizer cannot further increase the native mycorrhizal colonization rate.

## 4. Discussion

The essence of restoration of degraded grassland is an improvement of grassland production and ecological function, which are mainly manifested as the increase in vegetation productivity, soil fertility and microbial diversity. The results of this study showed that the grassland yield and soil nutrient contents of degraded artificial grassland can be increased by the application of organic fertilizer, indicating that it is an effective measure to restore degraded artificial grassland. The above results were consistent with previous studies by Han et al. [38] and Pan et al. [39], which were mainly attributed to the increase in available nutrient contents caused by organic fertilizer addition [40,41] and functional microorganisms enriched in the organic fertilizer [42,43]. The grassland yield and soil nutrient contents, especially contents of soil total nitrogen and nitrate nitrogen, significantly increased after testing the effectiveness of a grass–legume mixture inoculated with *Rhizobium* based on the application of organic fertilizer compared with CK. The symbiotic nitrogen fixation between leguminous forage and *Rhizobium* could be responsible for such an increase in yield and soil nutrients in degraded artificial grassland [44], indicating that the grass–legume mixture inoculated with *Rhizobium* can strengthen the positive impacts of the application of organic fertilizer on the yield and soil nutrient contents of degraded artificial grassland.

Soil microorganisms play an important role in nutrient cycling, and their activities can be easily affected by the addition of soil nutrients. The results of our study showed that the application of organic fertilizer increased soil bacterial diversity and fungal Chao 1 index, and it decreased the Simpson_evenness index of soil fungi. These alterations of the diversity of microbial communities were mainly due to the increase in soil organic matter content, which is rich in nutrients such as carbon, nitrogen and phosphorus, providing a good growth environment for microorganisms [45]. Changes in the root exudates of plants [46] and influences of external environment caused by the application of organic fertilizer are also important factors affecting microbial diversity [47,48]. However, the diversity of soil bacteria and fungi changed as a result of the grass–legume mixture or grass–legume mixture inoculated with *Rhizobium* based on the application of organic fertilizer, but the diversity was still higher than that of CK (no organic fertilizer), which was mainly attributed to changes in the soil physico-chemical properties caused by the grass–legume mixture [49], indicating that the grass–legume mixture inoculated with *Rhizobium* based on the application of organic fertilizer was a more effective measure maintaining the diversity of soil microorganisms than the application of organic fertilizer alone. Additionally, it was observed that the application of organic fertilizer significantly affected the community structure of soil bacteria and fungi, which is in agreement with previous studies [38,39,50]. However, the effects of the grass–legume mixture and grass–legume mixture inoculated with *Rhizobium* on the community structure of bacteria and fungi were not significant compared with the application of organic fertilizer. The above results might be related to changes of the soil C:N ratio and microbial metabolic function caused by organic fertilizer addition [51,52,53]. Furthermore, the taxonomic composition of soil bacteria and fungi at the levels of phylum and genus were also changed by applying organic fertilizer or the grass–legume mixture inoculated with *Rhizobium* based on the application of organic fertilizer. The abundance of dominant bacteria such as *Proteobacteria* and *Bacteroides* was increased by the application of organic fertilizer, while the abundance of *Chloroflexi* and *Actinobacteria* (*p* < 0.05) was significantly decreased; these changes were mainly attributed to the changes in vegetation and soil physico-chemical properties [54,55]. Moreover, the abundance of the dominant *Ascomycota* in the fungal community was also significantly decreased after the application of organic fertilizer (*p* < 0.05), as the majority of *Ascomycota* are saprophytic fungi [56], with some phytopathogenic, and the reduction in their abundance might be related to the decrease in pathogenic fungi caused by the application of organic fertilizer [57].

The findings of the Mantel test revealed that the content of soil nitrate nitrogen was the main driving factor for the alpha diversity of soil fungi (*p* < 0.05), while environmental factors had little effect on the structure, diversity, and species composition of the soil bacterial community, which were consistent with previous studies by Zhong et al. and Wang et al. [58,59], indicating that fungi may participate in the nitrification process of soil nitrogen. Moreover, soil fungi are important decomposing agents in ecosystems, participating in the decomposition and conversion of soil organic matter, and the diversity of fungi is closely related to soil fertility [60]. 

AMF are important biological components in soil, and soil fertility is considered a major determinant of the symbiotic efficiency of AMF. The results of our study showed that the colonization of native mycorrhizal fungi on the perennial gramineous plant in the degraded artificial grassland can be significantly improved by the application of organic fertilizer, which is in agreement with the view of Zhang et al. [61]. The increases in soil fertility and AMF spore density caused by organic fertilizer were the major reasons for the enhancement of the mycorrhizal colonization intensity of host plants [61,62]. However, the colonization rate of native mycorrhizal fungi decreased after conducting a grass–legume mixture or grass–legume mixture inoculated with *Rhizobium* based on the application of organic fertilizer, which was in agreement with the results of Buil et al. [63]. These results were mainly attributed to the increase in soil nutrients, and the increase in soil nutrients reduced the dependence of host plants on AMF. Meanwhile, previous studies have found a negative relationship between soil P availability and the richness and diversity of AMF [64,65]; the decrease in native mycorrhizal colonization in this study may be related to the increase in soil nutrient contents such as P and decrease in AMF abundance caused by agronomic measures such as the grass–legume mixture [66,67]. However, the effects of organic fertilizer on mycorrhizal fungi need further study.

## 5. Conclusions

In conclusion, the application of organic fertilizer can increase the yield and soil nutrient contents of degraded artificial grassland, and the effects can be further strengthened by using a grass–legume mixture inoculated with *Rhizobium* based on the application of organic fertilizer. The diversity, species composition, and community structure of soil bacteria and fungi can also be changed by these restoration measures, but the effects of the application of organic fertilizer were the highest. Moreover, the colonization of native mycorrhizal fungi on perennial gramineous plant can be significantly increased by the application of organic fertilizer, but it may be decreased by using a grass–legume mixture inoculated with *Rhizobium*. These findings can serve as a reference for the application of organic fertilizer in the ecological restoration of degraded vegetation and soil in alpine regions. However, the above results need further verification in different habitats and different growing seasons. 

## Figures and Tables

**Figure 1 microorganisms-11-01114-f001:**
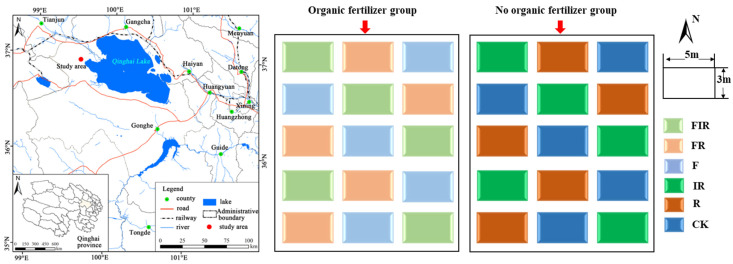
Schematic Diagram of Study Site and Design. F: Organic fertilizer; CK: No organic fertilizer + No seeding; FR: Organic fertilizer + Grass–legume mixture; R: Grass–legume mixture; FIR: Organic fertilizer + Grass–legume mixture inoculated with *Rhizobium*; IR: Grass–legume mixture inoculated with *Rhizobium*.

**Figure 2 microorganisms-11-01114-f002:**
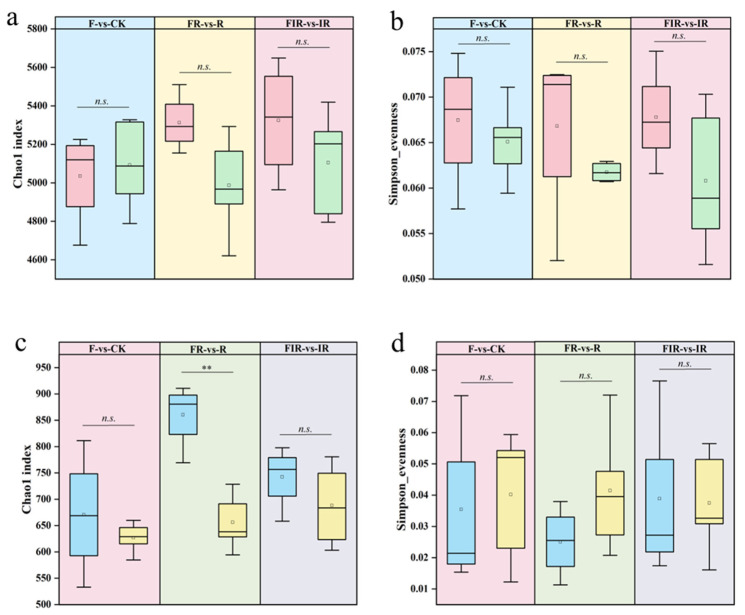
Alpha diversity of soil bacteria and fungi: Chao 1 index of soil bacteria (**a**) and fungi (**c**); Simpson_evenness index of soil bacteria (**b**) and fungi (**d**) (*n.s. p* > 0.05, ** *p* < 0.01).

**Figure 3 microorganisms-11-01114-f003:**
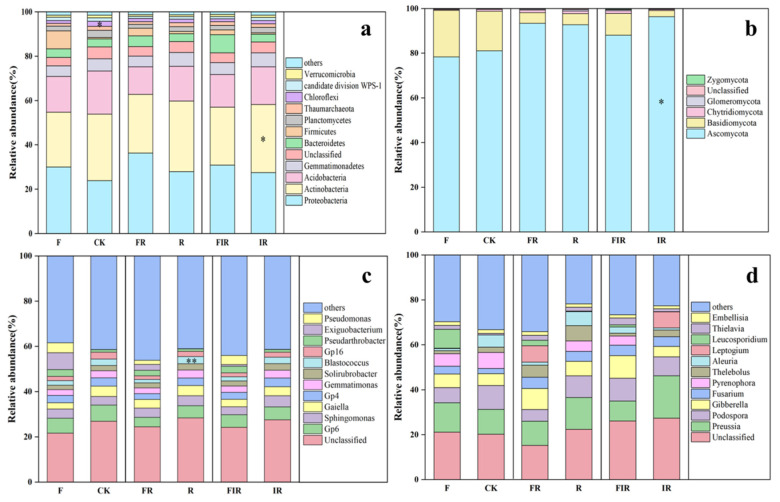
Taxonomic composition of soil bacteria and fungi: abundance of soil bacteria (**a**) and fungi (**b**) at the phylum level; abundance of soil bacteria (**c**) and fungi (**d**) at the genus level (* *p <* 0.05, ** *p <* 0.01).

**Figure 4 microorganisms-11-01114-f004:**
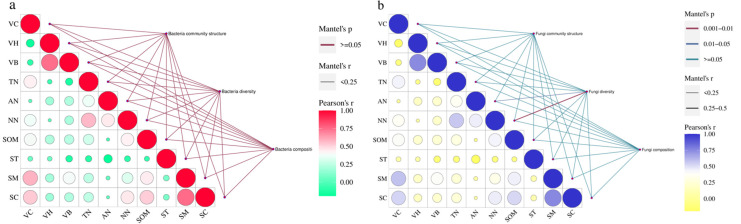
Charts of correlation between environmental factors and soil bacterial (**a**) and fungal communities (**b**). VC: Vegetation Coverage; VH: Vegetation Height; VB: Vegetation Aboveground Biomass; TN: Total Nitrogen; AN: Ammonia Nitrogen; NN: Nitrate Nitrogen; SOM: Soil Organic Matter; ST: Soil Temperature; SM: Soil Moisture; SC: Soil Conductivity. The size of the circle represents the degree of the correlation, and the correlation lines connect specific factors.

**Figure 5 microorganisms-11-01114-f005:**
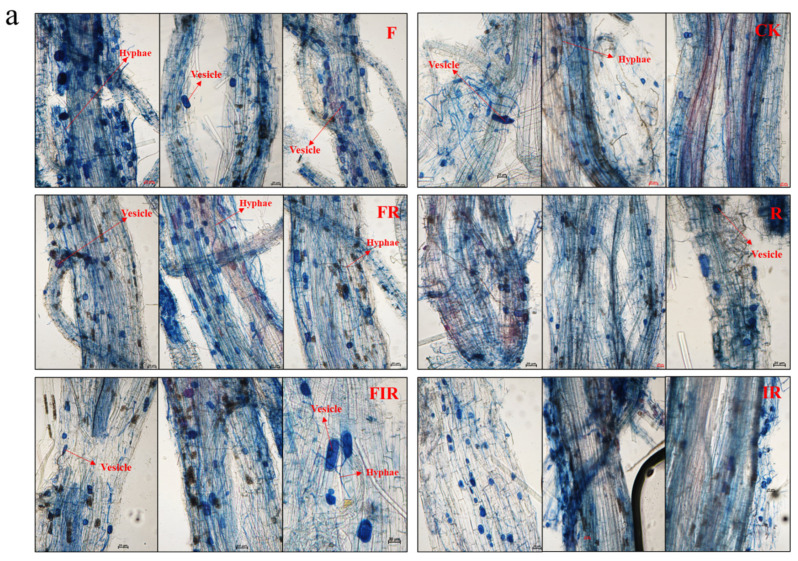
The native mycorrhizal structures (**a**) and mycorrhizal colonization rate (**b**) of the cortex of the *E. breviaristatus* root of each treatment. F: Organic fertilizer; CK: No organic fertilizer + No seeding; FR: Organic fertilizer + Grass–legume mixture; R: Grass–legume mixture; FIR: Organic fertilizer + Grass–legume mixture inoculated with *Rhizobium*; IR: Grass–legume mixture inoculated with *Rhizobium*. The more mycorrhizal structures in (**a**), the greater the mycorrhizal colonization rate (** *p* < 0.01, *** *p* < 0.001).

**Table 1 microorganisms-11-01114-t001:** Effects of the application of organic fertilizer and grass–legume mixture inoculated with *Rhizobium* on yield and soil nutrient contents of degraded artificial grassland.

Treatments	Vegetation Coverage/ %	Total Nitrogen/ mg/kg	Ammonia Nitrogen/ mg/kg	Nitrate Nitrogen/ mg/kg	Organic Matter/ %
F	85.0 ± 5.00 a	1647.42 ± 139.48 ab	4.82 ± 0.29 a	42.58 ± 3.40 bc	4.53 ± 0.44 ab
CK	57.0 ± 13.04 c	1404.72 ± 233.60 b	4.57 ± 0.61 a	42.17 ± 2.52 bc	3.52 ± 0.42 b
FR	77.0 ± 9.75 ab	1811.76 ± 287.35 a	4.36 ± 1.50 a	53.17 ± 7.22 ab	4.89 ± 0.42 a
R	68.0 ± 12.55 bc	1916.68 ± 260.97 a	6.43 ± 0.77 a	51.16 ± 1.12 abc	3.86 ± 1.04 ab
FIR	87.0 ± 4.47 a	1857.22 ± 196.34 a	5.59 ± 2.49 a	59.62 ± 12.41 a	4.65 ± 0.52 ab
IR	64.0 ± 13.99 bc	1641.72 ± 129.47 ab	4.72 ± 0.55 a	41.03 ± 1.42 c	3.82 ± 0.93 ab

Notes: F: Organic fertilizer; CK: No organic fertilizer + No seeding; FR: Organic fertilizer + Grass–legume mixture; R: Grass–legume mixture; FIR: Organic fertilizer + Grass–legume mixture inoculated with *Rhizobium*; IR: Grass–legume mixture inoculated with *Rhizobium*. Different lowercase letters indicate significant differences between treatments (LSD test, *p <* 0.05).

**Table 2 microorganisms-11-01114-t002:** PerMANOVA (Adonis) test of the effects of different treatments on the microbial community structure.

GroupFactors	Bacteria					Fungi				
	Df	SumOfSqs	F. Model	R^2^	Pr (>F)	Df	SumOfSqs	F. Model	R^2^	Pr (>F)
Application of Organic fertilizer	1	0.23561	2.5381	0.08311	0.018 *	1	0.3494	1.6509	0.05568	0.012 *
Complementary seeding	1	0.03831	0.3835	0.01351	0.858	1	0.1330	0.6248	0.02119	0.978
Inoculated with *Rhizobium*	1	0.05867	0.6096	0.02069	0.718	1	0.2588	1.2158	0.04124	0.174
Residuals	26	2.50226		0.88268		26	5.5344		0.88189	
Total	29	2.83484		1.00000		29	6.2756		1.00000	

Note: R^2^ represents the explanatory degree of GroupFactors to differences among samples, and the larger the value, the higher the explanatory degree. Pr represents the *p*-value, and *p* < 0.05 indicates the high reliability of this test (* *p* < 0.05).

## Data Availability

Our data were deposited in the China National Microbiology Data Center (NMDC) with accession numbers SUB1670298073368. The relevant data are available from the corresponding author on request.

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
