# Peer review of "Grass–Legume Mixture with Rhizobium Inoculation Enhanced the Restoration Effects of Organic Fertilizer"

_microorganisms, 2023, doi:10.3390/microorganisms11051114_

Round 1

Reviewer 1 Report (Previous Reviewer 1)

This publication presents interesting results related to the effects of the organic fertilizers, planting the grass-legume mixture and Rhizobium inoculation on yield, soil nutrient contents, and microbe’s community composition in artificial degraded grassland. The adopted approach in this study was interesting since it could provide more insights into the use of organic fertilizer and grass-legume mixture in the ecological restoration of degraded grassland.

The manuscript was well introduced, and the authors adopted methods with discussion of the different obtained results. However, the manuscript needs major revisions to be suitable for publication in Microorganims.

General comments

- Comment 1: The English of this manuscript needs some improvements.

- Comment 2: The discussion section should be more developed.

Other comments

- M&M

L126: please delete the space between the value and the %. Please check throughout the manuscript. We add a space the unit instead of the %.

L162-170: please add reference for each used method.

- Discussion

L392: please change “its” to “their”.

Please provide more detailed discussion of your results, especially the effect of the different applied treatments on native mycorrhizal colonization rate.

Author Response

Reviewer 2 Report (New Reviewer)

The manuscript microorganisms-2248898, entitled “Grass-legume Mixture with Rhizobium Inoculation Enhanced the Restoration Effects of Organic Fertilizer” submitted by Zhang et al. reported and discussed the results from a one year field experiment where the effect of grass-legume mixture with and without Rhizobium inoculation and with and without organic fertilization were assessed. Considering the importance to study strategy to improve soil fertility and productivity of natural grassland, also by enhancing its microbioma, I believe that the manuscript is of potential interest to readers of “microorganisms” and falls within its scope.

In general, the experimental activity was carried out following strict scientific logic and according to widely used methods which have made it possible to obtain reliable results.

In my opinion, the manuscript needs only some minor revisions described in the attached pdf file.

Abstract: Well written.

Keywords: Revised according to my specific suggestion in the enclosed pdf file. Try to do not write words already mentioned in the manuscript’s title.

Introduction: Small changes are needed in order to improve content and style.

Materials and Methods: In general this part is fine. Consider to simply some parts and add some other details according to my suggestion in the pdf file.

Results: well written and fluent. Only minor changes are needed.

Discussion: in general, is clear and well-written.

Conclusion: clear, and well-written and summarize the main results observed in the experiment.

My specific comments, which I hope will help the authors to improve their manuscript, are enclosed in the attached pdf file.

Author Response

This manuscript is a resubmission of an earlier submission. The following is a list of the peer review reports and author responses from that submission.

Round 1

Reviewer 1 Report

This publication presents interesting results related to the effects of organic fertilizers, planting the grass-legume mixture, and Rhizobium inoculation on yield, soil nutrient contents, and microbe community composition in artificial degraded grassland. The adopted approach in this study was interesting since it could provide more insights into the use of organic fertilizer and grass-legume mixture in the ecological restoration of degraded grassland.

The manuscript was well introduced, and the authors adopted methods with the discussion of the different obtained results. However, the manuscript needs major revisions to be suitable for publication in Microorganims.

General comments

- Comment 1: The English of this manuscript needs some improvements.

- Comment 2: The illustration captions should be more specific.

- Comment 3: The discussion section should be more developed.

Other comments

Title: It could be better to include the use of the grass-legume mixture and Rhizobium inoculation in the title.

- Abstract

Keywords: please add “Rhizobium inoculation”.

- M&M

L106: please add a space between the value and the unit. Please check throughout the manuscript.

L120-122: This study would have been more reliable if it had been repeated for a second season or conducted at the same time in two different pedoclimatic regions.

L126-129: Did the organic fertilizer contain any microorganisms? if yes, which are? If not, please mention that it was sterilized.

L238: please add a dot after “(p < 0.05)”.

- Results

L274: please be more specific by adding the other factors used besides organic fertilizer. The same for all the other illustration captions.

L285: please change “increased significantly” to “significantly increased”. Please check throughout the manuscript.

- Discussion

Please provide a more detailed discussion of your results, especially the effect of the different applied treatments on the soil microbe’s community and structure and the correlation analysis.

L417-419: Please explain.

- References

Please provide the same format of references. For instance, the journal name was written sometimes in full and sometimes in abbreviation.

All species' scientific names should be italicized.

Reviewer 2 Report

The manuscript title "Organic Fertilizer Switched Soil Microbial Community and Enhanced Native Mycorrhizal Colonization in Artificial Grass-land " covers important information for readers. However, it needs a major revision at the present stage.
1)The title is not addressing the output of the study, and I would like to see a revised title that should also be stated what this community shift works and its vitality.
2) The Introduction part looks very basic, and it needs to be more focused as it is an important analysis/method that has often impacted negatively the crop and how both kinds of communities can complement each other without competing with each other. …these all information may make the introduction more interesting.
3) The method part is not written in detail. Kindly describes all the method that use in brief that will help to repeat this analysis and also add the flow chart to Figure 1 of methods used (step by step).
4). The manuscript needs minor scientific editing.
5) Author should need to discuss the results more focused way as per the hypothesis and results. In the manuscript discussion is very poor, the author should revise the whole discussion part and add the future perspective of the study.

Reviewer 3 Report

Please address the following comments before any decision to consider this paper for publication.

- The first half of the title is not appropriate and do not give clear meaning. Please write/rephrase it properly, so that it can be understandable.

- L24-26, while reporting results, authors cannot use 'could'. Please report in a direct way, so that it should reflect anything as concrete what your found. Same is suggested for L27

- The results section reported in Abstract is not in a proper way, merely general statements. Your results could not be just as general statements. Be specific, that how much increase you had observed through Organic fertilizer.

- What are the reasons of grassland degradation??? please mention in Introduction section.

- Quote examples from previous research about artificial grasslands

- The introduction is too lengthy, be brief and specific about the study problem.

- L94-97, please rewrite this objective statement, to give clear meaning. You may break the sentence here.

- L165 to L171, please provide full details about the sample preparation, handling and the equipment used to measure the soil N and other nutrients contents., same is recommended for rest of the methods section.

- L241-243, please correct this sentence for Grammar.

- WHy the authors used Oneway ANOVA? why not you used 2-way ANOVA? didn't you study interactions?

- L251, its understood that any soil input will case change in yield and nutrients, remove this sentence. and do brief reporting for results section.

- What is CK??? do proper coding for treatments.

- Table 1, spell out all the abbreviations/treatment codes used in each table as footnote, so that each table/figure should be self-explanatory.

- L377, Write this sentence in correct English.

- L387-391, Authors should be sure enough while discussing results, for this a thorough literature review is needed, as lot of studies have been conducted so far to study effect of OM in soil microbial communities.

- The discussion section is week.

- Avoid self-citing your works, currently 7 papers have been cited here.

Round 2

Reviewer 1 Report

The authors have satisfied all the raised comments and the manuscript is now suitable for publication.

Author Response

Thank you again for your valuable comments and suggestions on my article!

Reviewer 2 Report

The manuscript Improved as per my suggestions.

Author Response

Thank you again for your valuable comments and suggestions on my article

Reviewer 3 Report

The authors have not addressed the comments and suggestions carefully in the revised version. The discussion section is not improved in the revised version and also there are a lot of grammatical and spelling mistakes in it. Please revise and resubmit.

Author Response

Thank you again for your precious comments and suggestions on my article, and I am very sorry that I had not addressed the comments and suggestions carefully in the first revised version. This time we revised our article in detail including the discussion section, at the same time, we corrected the grammatical and spelling mistakes in the article and marked it in a yellow background color.
